# The Potential of using Remote Sensing data to estimate Air-Sea $CO_2$ exchange in the Baltic Sea.

Gaëlle Parard[1,3], Anna Rutgersson[1], Sindu Raj Parampil[1], and Anastase Alexandre Charantonis[2]

[1]Department of Earth Sciences, Uppsala University, Uppsala, Sweden,
[2]École nationale supérieure d'informatique pour l'industrie et l'entreprise, Evry, France
[3]AGO-GHER-MARE, University of Liège, Allée du Six Aout, 17, Sart Tilman, Liège 4000, Belgium

*Correspondence to:* Gaëlle Parard (parard.gaelle@gmail.com)

**Abstract.** In this article, we present the first climatological map of air–sea $CO_2$ flux over the Baltic Sea, based on remote-sensing data: satellite imaging derived estimates of $pCO_2$ using self-organizing maps classifications along with class-specific linear regressions (SOMLO methodology) and remote-sensed wind estimates. The estimates have a spatial resolution of 4-km both in latitude and longitude and a monthly temporal resolution from 1998 to 2011. The $CO_2$ fluxes are estimated using two types of wind products, i.e. reanalysis winds and satellite wind products, the higher-resolution wind product generally leading to higher-amplitude fluxes estimations.

Furthermore, the $CO_2$ fluxes were also estimated using two methods: the method of Wanninkhof et al. (2013) and the method of Rutgersson and Smedman (2009). The seasonal variation in fluxes reflects the seasonal variation in $pCO_2$ unvaryingly over the whole Baltic Sea, with high winter $CO_2$ emissions and high $pCO_2$ uptakes. All basins act as a source for the atmosphere, with a higher degree of emission in the southern regions (mean source of 1.6 mmol m$^{-2}$ d$^{-1}$ for the South Basin and 0.9 for the Central Basin) than in the northern regions (mean source of 0.1 mmol m$^{-2}$ d$^{-1}$) and the coastal areas act as a larger sink (annual uptake of -4.2 mmol m$^{-2}$ d$^{-1}$) than does the open sea (-4 mmol m$^{-2}$ d$^{-1}$). In its entirety, the Baltic Sea acts as a small source of 1.2 mmol m$^{-2}$ d$^{-1}$ on average and this annual uptake has increased from 1998 to 2012.

ir-sea $CO_2$ flux, Baltic Sea, neural method, climatology.

## 1 Introduction

From the early 2000 and onwards, there has been a more active attempt to investigate, understand, and quantify the global carbon cycle by the scientific community, since the greenhouse gas carbon dioxide ($CO_2$) plays a key role in controlling Earth's climate. The oceanic uptake of anthropogenic $CO_2$ helps regulate atmospheric $CO_2$ through air–sea exchange. Coastal and marginal seas represent nutrient-rich areas with strong biological activity and are influenced by various anthropogenic factors. As the oceans take up a major part of the anthropogenic emissions of $CO_2$, many oceanic regions are experiencing ongoing acidification. There are still major uncertainties in assessing the oceanic uptake of anthropogenic $CO_2$: during 2005–2014 it was estimated to 2.6 GtC yr$^{-1}$, an estimated 26% of the total anthropogenic $CO_2$ emissions (Le Quéré et al., 2015). One reason for this uncertainty is the lack of reliable information on the coastal seas, which have so far barely been considered in the oceanic and global carbon budgets. The coastal ocean's role in terms of carbon export and relative productivity is disproportionately

large in respect to its total surface area (7%), when compared with the open ocean (Bourgeois et al., 2016). As the annual amplitude of air–sea $pCO_2$ difference is significantly larger in coastal regions than open ocean(Rödenbeck et al., 2013), the variability of the exchange is high. Several study show the

Various methods, both direct and indirect, are used to determine the air–sea flux of $CO_2$ ($FCO_2$) (e.g. Smith et al., 1996; McGillis et al., 2001; Krasakopoulou et al., 2009). Both direct and indirect measures of $FCO_2$ were used in this study (McGillis et al., 2001; Rutgersson and Smedman, 2009; Gutiérrez-Loza and Ocampo-Torres, 2016).

Other studies have calculated $FCO_2$ across ocean basins using climate databases (Takahashi et al., 2002) or biogeochemical numerical models (Lenton et al., 2013; Arruda et al., 2015). These calculations, however, have failed to provide outputs covering the global coastlines. This is primarily due to the sparseness of the temporal and spatial data-sets (such as $pCO_2$ of the surface ocean or wind fields). The wide range of values of in situ coastal $FCO_2$ entails even wider uncertainties in global estimates of $FCO_2$, as there is the potential to under- or overestimate $FCO_2$ when performing a spatio-temporal integration (Wollast, 1991; Takahashi et al., 2009; Ribas-Ribas et al., 2011). A better comprehension of the local processes controlling $FCO_2$ along each coastal setting of continental margins will therefore lead to a better constrained set of global $FCO_2$ estimates. Since the year 2000, many different $FCO_2$ estimates and measurements have been reported for various near-shore, coastal, and inner-shelf environments. The question about the coastal seas which can be a source or a sink remained open until recently, in Chen et al. (2013) the coastal sea act as a sink with a mean value of air to sea flux is -1.09±2.9 mol C $m^{-2}yr^{-1}$. The study show that most of the shelves absorb $CO^2$ from the atmosphere except at the low latitudes where they act as a source (0.11 Pg C $yr^{-1}$) compare to high and temperate latitude (-0.33 pG C $yr^{-1}$. The study show that the shelves in the Atlantic Ocean have the highest total absorption which represent 33% of the total of the absorption which represent a mean air sea $CO_2$ flux of -1.2mol C $m^{-2}$ $d^{-1}$. The spread of these values is a result of the heterogeneous and coupled biogeochemical processes in near-shore and coastal systems (Laruelle et al., 2010). It is necessary to increase our comprehension of the ocean carbon cycle and the air–sea exchange of $CO_2$ along the continental margins (Alin et al., 2012), due to their high social and ecological impact (Vargas et al., 2012).

High biological activity causes high $CO_2$ fluxes between the coastal and marginal seas and between the atmosphere and adjacent open oceans, respectively. Considering their combined surface area, coastal seas may contribute disproportionately to the open-ocean storage of $CO_2$ (Thomas et al., 2004) via a mechanism called the continental shelf pump (Tsunogai et al., 1999). In recent years, detailed field studies of $CO_2$ fluxes have been initiated in a few areas, such as the East China Sea, Northwest European Shelf, Baltic Sea, and North Sea (Chen and Wang, 1999; Thomas et al., 1999; Thomas and Schneider, 1999; Frankignoulle and Borges, 2001; Borges and Frankignoulle, 2002; Borges et al., 2003; Thomas et al., 2003, 2004).However, only limited information is available on a global scale about these $CO_2$ fluxes (Liu et al., 2000a, b; Cai et al., 2003; Chen et al., 2003; Omstedt et al., 2009; Norman et al., 2013b).

The Baltic Sea is a semi-enclosed sea in Northern Europe (Meier et al., 2014) which has been relatively well studied (e.g. Omstedt et al., 2004; Hjalmarsson et al., 2008; Backer and Leppänen, 2008; Wesslander, 2011) and monitored, and can be used in developing new methods for monitoring coastal seas. It is characterized by river runoffs (Bergstrom, 1994) which are 2015 estimated 17241.9 $m^3$. $s^{-1}$ (Johansson, 2017) as well as by an important upwelling variability (Norman et al., 2013a; Myrberg

and Andrejev, 2003; Lehmann and Myrberg, 2008; Sproson and Sahlée, 2014). In the Baltic Sea, (Siegel and Gerth, 2012) shows that decomposition of organic matter and biological production control the biogeochemical processes. The nutrient and carbon distribution in the water column, as well as light availability are the limiting factors of these processes. In the Baltic sea, the former factors are affected by physical constraints such as the stratification of the water, the salinity and temperature profiles as well as the sea currents.

In recent years, the Baltic Sea has also been paid more attention as a coastal system affecting both the uptake/release of anthropogenic $CO_2$ and the natural $CO_2$ cycle (Thomas and Schneider, 1999; Lansøet al., 2015). Between 1994 and 2008 direct $CO_2$ measurements from a cargo ship has been recorded, with a monthly resolution. The net annual air–sea exchange of $CO_2$ in the central Baltic Sea and the Kattegat varied both regionally and inter-annually. In the examined period, the Kattegat sea was, on average, a sink of $CO_2$ while the East Gotland and Bornholm seas were sources. The air–sea exchange of $CO_2$ and gas transfer velocity interannual variations were more pronounced in winter periods than in the summer periods. This indicates the interannual variability in the annual net flux is mainly controlled by the winter conditions (Wesslander et al., 2010).

The balance between mineralization and production, as well as the depth of the mixed-layer in the different oceanic zones examined were shown to be the main drivers of their respective sink / source distributions (Wesslander et al., 2010). In the central Baltic Sea, $CO_2$-enriched water mixes with water up to the surface in winter. The central Baltic Sea also receives large amounts of organic material from river water inflow; this may give rise to a heterotrophic system, making the central Baltic a net $CO_2$ source. This is not the case in the Kattegat, which is highly influenced by oceanic conditions.

In this study, the air sea $CO_2$ flux is estimated, with the ocean-surface $pCO_2$ in the Baltic Sea estimate from satellite-data derived products in (Parard et al., 2015, 2016). The outputs of the method have a horizontal resolution of 4 km and cover the period from 1998 to 2011. Previous studies of the net uptake or release of $CO_2$ in the Baltic Sea have produced a wide range of results, with net exchange varying between –3.6 and +2.9 mol $CO_2$ m$^{-2}$ y$^{-1}$ in different time periods between 1994 and 2009 (Norman et al., 2013b).

The goal of the present study is to develop an air–sea $CO_2$ flux estimation based on remote-sensing products with a monthly time resolution and 4° spatial resolution and to estimate the error of this method of flux estimation in the Baltic Sea. In addition, we will further describe the processes and air–sea fluxes of $CO_2$ from 1998 to 2011 in the entire Baltic Sea and discuss the advantage and the limit of the method

The study is structured in four sections. Section 2 presents the data and method used in this work. Section 3 presents the wind products used to estimate the exchange (based on satellite data and reanalysis data). In Section 4, we analyze the wind products' quality, as well as various aspects of the estimated fluxes , and in Section 5 we present our conclusions.

## 2  Data and method

### 2.1  pCO$_2$ map

We used the SOMLO methodology (Sasse et al., 2013), to reconstruct the sea-surface $pCO_2$ concentrations. The SOMLO methodology combines two statistical approaches: *self-organizing maps* (SOMs) (Kohonen, 1990) and *linear regression*.

SOMs are a subfamily of neural network algorithms used to perform multidimensional classification. During its training phase, the SOMLO methodology first uses SOMs to discretize a dataset of explanatory parameters into classes and then locally learns a set of linear regression coefficients to infer the $pCO_2$ for each class. When presented with a new vector of explanatory parameters, it first classifies it on the SOM map, then uses the calculated regression coefficients to estimate the $pCO_2$.

We divided the Baltic Sea (BS) into four regions in (Parard et al., 2016): the Gulf of Bothnia (GB), Gulf of Finland (GF), Central Basin (CB), and South Basin (SB) (Fig. 1).

We then trained the SOMLO methodology on the data belonging to each of these basins, reconstructing each point by combining the results obtained through each training, weighted by the distance from each point to the center of each region.

The covariance of the explicative variables with the $pCO_2$ was taken into account when attributing a data vector to a class, by means of a modified distance function. This allows for certain extreme parameter values to be more easily associated with the areas of the SOM where the $pCO_2$ is more correlated with these values.

In addition, we chose to perform a principal component analysis (Jolliffe, 2002) of the training data belonging to each class of each SOM. We kept the first four axes of the principal component analysis and taught the regression coefficients using the data projections on these four axes instead of performing a regression on all the parameters.

## 2.2 Wind products

In this study we used wind products to calculate the transfer velocity, based on a meso-scale reanalysis product. A reanalysis is a combination of measurements and a model in which the available data are assimilated into a high-quality numeric modeling system. The reanalysis used in this paper was provided by the Swedish Meteorological and Hydrological Institute (SMHI) with the High-Resolution-Limited Area Model (HIRLAM) geometry (22-km horizontal grid spacing and 60 levels in the vertical; the model top is at 10 hPa) (Soci et al., 2011) . HIRLAM is downscaled and dynamically adapted to a higher resolution (5-km grid) with a simplified HIRLAM called the Dynamic Adaptation Model (DYNAM). The observations of 10-m winds assimilated into the system are from four databases: the Integrated Surface Database Station History (ISH) database maintained by NOAA's National Climatic Data Center (NCDC), the MARS archive at ECMWF, the European Climate Assessment & Dataset (ECA&D) used as input for E-OBS version 6.0, and the national climate databases of SMHI and Météo France (MF). The temporal resolution is of 6 hours. In the following, this product will be referred to as SMHIp. The method requires for the explicative data to stay coherent in terms of resolution, and as such we chose a temporal and spatial resolution of monthly, 4 x 4 km $pCO_2$ pixels.

In order to estimate the impact of the wind product on the air-sea $CO_2$ flux, we computed the flux with a remote sensing product at daily scale. The wind data are reprocessed QuikSCAT (QSCAT) and ASCAT data (Bentamy and Croizé-Fillon, 2013) with a spatial resolution of 25x 25 km. The data are available from 2000 to 2011.

## 2.3 Calculation of CO$_2$ flux

The flux of $CO_2$ (FCO$_2$) from sea to air (positive value) or air to sea (negative value) is often calculated using the difference in the partial pressure of $CO_2$ between the surface water and the atmosphere ($\Delta pCO_2$).

Here, the atmospheric $pCO_2$ was estimated using the method from Rutgersson et al. (2009) and the sea-surface $pCO_2$ concentrations are reconstructed with the SOMLO methodology (Sasse et al., 2013), as done by Parard et al. (2015, 2016). The SOMLO methodology combines two statistical approaches: *self-organizing maps* (SOMs) (Kohonen, 1990) and *linear regression*.

In addition, the exchange efficiency was required, which was expressed in terms of a transfer velocity, $k$. The flux was then calculated according to:

$$FCO_2 = kK_0 \Delta pCO_2 \tag{1}$$

where $K_0$ is the salinity- and temperature-dependent solubility constant (Weiss et al., 1982). The gas transfer velocity was computed using the parameterization from (Wanninkhof et al., 2009):

$$k = \sqrt{\frac{660}{Sc}}(3 + 0.1U + 0.064U^2 + 0.011U^3) \tag{2}$$

where $U$ is the wind velocity at a reference height of 10 m and $Sc$ is the solubility-dependent Schmidt number. Daily values of $k$ were computed with a 6-h frequency for SMHIp; Eq. 2 is valid for all wind speed ranges. This method will be define as Method 1.

We compare the results with another method to compute the transfer velocity k from Rutgersson and Smedman (2009)

$$k = 0.24 * U^2 + (3022 * w - 20) \tag{3}$$

where $w$ is the water-side convection this is estimated from the model used in Norman et al. (2013b). This method will be defined as Method 2 .

## 3 Results

### 3.1 Analysis of the wind products

#### 3.1.1 Validation of the wind product

To validate our wind product, we compare the SMHI product with one based on remote-sensing data at daily scale 10 m wind data are reprocessed QuikSCAT (QSCAT) and ASCAT data (Bentamy and Croizé-Fillon, 2013) with a spatial resolution of 25x 25 km here called SATp. The two products are quite coherent when compared to all the station data used here, though SMHIp seems better, having a higher average correlation coefficient, i.e. $R = 0.84$ versus 0.67 for the remote sensing data wind (we chose not to show here). This is to be expected, as SATp has a much coarser spatial resolution (25 km) than SMHIp does (5 km). In the following we decided to used the SMHI product to compute the transfer velocity.

The wind product SMHIp used here to compute the air–sea $CO_2$ flux was compared with wind-tower data available from 24 stations in the Baltic Sea, including data from the Östergarnsholm measurement site Högström (2008); Rutgersson et al.

(2008). Here, a micro-meteorological tower, situated at 57.42°N, 18.99°E, has been running since 1995, making high-quality wind speed measurements at five heights. To validate the satellite data, we used measurements made 12 m above mean sea level in the 1995–2002 and 2005–2009 periods. In addition, we validated the winds using synoptic station data from SMHI for 21 sites along the coast of Sweden.

The wind product SMHIp agree quite well with the station data (Table 1). Most of the synoptic stations are very close to the coast, so there might be a bias due to land influence. The correlation coefficient ($R$) is quite high (0.66–0.91).

The root-mean-square differences (RMSDs) is given in Table 1.

The SMHIp have a quite high average correlation coefficient, i.e. $R = 0.84$ (Table 1). This is to be expected given that the spatial resolution is quite high for SMHIp (5 km).

We increase the resolution of the wind products by means of linear interpolation to compute the air–sea $CO_2$ flux. This was done to provide coherency between our datasets.

### 3.1.2   Wind variability over the Baltic Sea.

We examine the annual and monthly mean wind speeds and wind variability for the entire Baltic Sea (Figs. 2) for the twelve month during 13 years from 1998 to 2011. Fig. 2 shows the wind speed in colors and the annual wind variability in contours at

the seasonal time scale. The mean winds are higher in the Central Basin (CB) than the Gulf of Bothnia (GB), i.e. about 7–7.4 m s$^{-1}$ versus 5–6 m s$^{-1}$. The wind pattern agrees qualitatively with those in previous studies. In terms of variability, the wind can vary by as much as 1.5–2.1 m s$^{-1}$ in both CB and 1.4-1.9 m s$^{-1}$ in GB. On the monthly scale, high mean winds (8–9 m s$^{-1}$) are seen in the Baltic Sea from November to February (Fig. 2). Of the four regions, CB experiences the highest winds in winter months. March and September are transition months with winds generally between 7 and 8 m s$^{-1}$. May and June are

the months when the winds are generally low, 4–5 m s$^{-1}$. The largest variability in the winds, as represented by the contours (Figure 2), is observable from September to December. The variability remains strong from December to February (1.2 -2.4 m s$^{-1}$) in all the basins, while the lowest variability is observed in July (< 0.8 m s$^{-1}$).

### 3.2   Air–sea $CO_2$ flux

### 3.2.1   Air-sea $CO_2$ flux estimation and variability

The air–sea $CO_2$ flux estimations are shown in Figure 3, fluxes are computed using the SMHIp wind data and figures represent the time period from 1998 to 2011. Figures 3 and 4 show the seasonal cycle, we observed the same patterns reflecting the surface p$CO_2$ partial pressure (the air-sea difference in partial pressure) previously seen in (Parard et al., 2016). April to August represents an uptake and October to February an outgassing. The interannual variability is slightly larger during the spring, this can indicate a large interannual variability on the onset of biological activities. Spatial differences are larger during the

biologically active period. For example, in April the northern basins act as a source areas while the southern basins represents an uptake of the atmospheric $CO_2$. Transfer velocity is largest in the southern basin and during winter following the wind-speed pattern. In Figure 4, the annual mean concentrations are shown. The flux displays high seasonal and spatial variability, ranging

from –11 to 27 mmol m$^{-2}$ d$^{-1}$. On average, between 1998 to 2011, the entire Baltic Sea acts as a sink of –1.2 mmol m$^{-2}$ d$^{-1}$ (Figure 3). The values estimated from the remote sensing products are in agreement with those from other studies, indicating that the Baltic Sea can be a small source on average or a small sink of $CO_2$. Most previous research results concerning the carbon budget cover shorter periods, indicating a range between -1.16 and 2.9 mol m$^{-2}$ y$^{-1}$ (Wesslander, 2011; Kulinski and Pempkowiak, 2012, e.g.), though the maximum values reported in these studies are all found in the same one or two years (Algesten et al., 2006). Half of the studies demonstrate that Baltic Sea or certain basins of it act as sources, while the others demonstrate that it acts as a sink for the atmosphere (Norman et al., 2013a). In (Chen et al., 2013), the Baltic Sea show a air-sea $CO_2$ flux of -1.95 mol m$^{-2}$ yr$^{-1}$ which is also in agreement with the result of our method.

The Baltic Sea is divided into four regions; the annual mean values for transfer velocity, pCO$_2$ and fluxes for these four regions are presented in Fig 4.

During all the study period, the four basin acts in general as a source. The Central Basin acts as a source except for 4 years 2003, 2004, 2009 and 2010 with a lower value in 2009(—0.8 mmol m$^{-2}$ d$^{-1}$). The Gulf of Finland acts as a source of the same order of magnitude as the Central Basin with 4 years as a sink 2005,2007,2008 and 2009 with a lower value in 2009(—0.8 mmol m$^{-2}$ d$^{-1}$). The South Basin and the Gulf of Botnian acts as a source in all the years except respectively 2010 with a low sink (—0.01 mmol m$^{-2}$ d$^{-1}$) and 2009 (-0.4 mmol m$^{-2}$ d$^{-1}$). The interannual variability is the same order of magnitude for all the basins however the largest variability is seen in the Gulf of Bothnia, acting as a source until 2008 (>1.7 mmol m$^{-2}$ d$^{-1}$) and a smaller source afterwards (< 0.8 mmol m$^{-2}$ d$^{-1}$). The seasonal cycle do not show different patterns for the different basins. The seasonal cycle is smaller for the northernmost basin (GB) (Figure 3).

Between 1998 and 2011, the annual air–sea $CO_2$ flux in the Baltic Sea is always positive (Figure 4) but we observed higher flux before 2003 and after 2007. The four basins display a decrease in the flux from 1998 to 2011 (Figure 4). The decrease is larger in the Gulf of Bothnia, after 2008 the value are less than the half than the value before. A smaller decrease is observed in the Gulf of Finland. A decreasing trend can be explained by transfer velocity or pCO$_2$, but the decreasing pattern in the flux is not really reflected in the annual values of these parameters. The trend can also be explained by changes in seasonal distribution of parameters. The seasonal cycle shows a shift in time when comparing the first five years (1998 to 2002) compared to the last five years (2007 to 2011) in Figure 5. In all the basins the uptake is larger in April and May. For the later period, the differences is particularly large in the basins most influenced by ice cover (GB and GF). There is also an indication in GB and GF for a reduced outgassing in early winter. As the data is not entirely homogeneous as it describe in Parard et al. (2015) one should not draw too far conclusions from the suggested trend. It could, however, be related to the higher pCO$_2$ concentrations in the atmosphere due to anthropogenic emissions, the corresponding increase in $CO_2$ concentration in the atmosphere during this period is 23.7 $\mu$atm. As the trend to a large extent is explained by an earlier onset of spring-time uptake differences in temperature and ice cover might be a more likely explanation.

The coastal region is defined by a distance of 0.5° in latitude and longitude from the coast. Farther than 0.5° in latitude and longitude from the closest coast is defined as the open sea. The $CO_2$ flux compute in the coastal region is lower in winter and higher in summer than it is in the open sea (Fig. 6). The average difference in $CO_2$ flux is –0.5 mmol m$^{-2}$ d$^{-1}$ with a variability of between -5.5 and 2.5 mmol m$^{-2}$ d$^{-1}$. The higher difference (-1.6 mmol m$^{-2}$ d$^{-1}$) is observed in 2007 with a lower value

for coastal region. The air-sea $CO_2$ fluxes are lower for all the year in the coastal region. Annually, there are three periods when we observe a greater difference, i.e. February–March, June–July, and October (Fig. 6). The biological activity is one of an explanation of the lower air-sea $CO_2$ in the coastal region in March–April and October compare to the open ocean region. The biological activity is higher along the coast at these times (Schneider, 2011) due to upwelling near the coast (Omstedt et al., 2009; Norman et al., 2013a); this has the effect of reducing the $CO_2$ emitted to the atmosphere. In the coastal region we observed a change in the sink between the first five years between 1998 and 2002 and the last five years between 2007 and 2011 (Figure 7), The lower air-sea $CO_2$ flux are observed during the last years and the the minimum of the air-sea $CO_2$ flux is in April and May. It is correlate with the observation in the Figure 5. The sink increase in April from -2.9 mmol m$^{-2}$ d$^{-1}$ and in May from - 1.8 mmol m$^{-2}$ d$^{-1}$. The monthly difference is small compared with that observed at the seasonal scale, though we may be underestimating the effect of the upwelling at the monthly scale. A review of Baltic Sea upwelling (Lehmann and Myrberg, 2008) demonstrates that the typical upwelling lasts from several days to one month at a horizontal scale of 10–20 km offshore. It is therefore possible that the effect of the upwelling may be underestimated.

### 3.2.2 Uncertainty analysis

The method used to compute the $pCO_2$ has an advantage to compute a monthly map $pCO_2$ map for the entire Baltic Sea from 1998 to 2011 a data set of in situ data present in figure 1. As it explain in (Parard et al., 2016) for the reconstructed $pCO_2$ values The correlation coefficient ($R$) values are good, the lowest being observed in the Southern Basin (0.9) where the RMS is the highest (i.e., 38.5 $\mu$atm). The Gulf of Finland has the highest $R$ value (i.e., 0.97) and the Gulf of Bothnia the lowest RMS (19.5 $\mu$atm), the latter being the region with the lowest data density. This error have an impact on the air-sea $CO_2$ flux computation. The impact of the maximum RMS on the flux is $\pm$ 4 mmol m$^{-2}$ d$^{-1}$. This give a high influence of the air-sea $CO_2$ flux and our incertitude on the air $pCO_2$ increase this incertitude.

The difference between the phase before 2003 and after 2007 could be explained by the repartition of the data used to calculate our results. In order to understand if this repartition of the initial data is responsible for the phase difference, we studied the representation of the data along the different years for each neuron of the SOM maps in each basin (Figure 8). For the three first basins (Figure 8,a.,b.,c.), all the years are present at least in part, even if some classes seem to be solely composed from data measured before 2002, in particular in the Southern regions (the blue trend color classes). In the North of the Gulf of Bothnian there is no data before 2008 so the results that we show can be affected by this lack of data, yet is coherent with the other basins. The distribution of the data is well spread (Figure 8,e.,f.,g.,h.) throughout the classes.

Two tests were performed in order to estimate the error on the air-sea $CO_2$ flux. One with SATp wind product and one with the air-sea flux estimations method Rutgersson et al. (2009) describe in eq. 3. These results are presented in Figure 9. The two air–sea $CO_2$ flux estimations are computed using the two sets of wind data, the SMHIp and SATp datasets. The $CO_2$ flux computed using SMHIp wind data is available from 1998 to 2011 and using SATp wind data from 2000 to 2011. We compared the two products from 2000 and 2011 (Not show here). the two flux estimations from the wind product have the same order of magnitude. Nevertheless, the seasonal cycle from air-sea $CO_2$ flux using SATp product is larger, with lower value in summer and higher in winter. We observe the maximum difference in January (when the flux using SMHIp winds is higher) and in

September (when the flux using SATp winds is higher). The monthly variability of the flux using SMHIp winds is 8.7-11.4 mmol m$^{-2}$ d$^{-1}$ versus 3.4-13.4 mmol m$^{-2}$ d$^{-1}$ using SATp winds. High variability in January using the SATp wind product can be explained by the lack of satellite data during for this month. In addition, there are also interannual variations. In most years, the Baltic Sea acts as a sink: using the SMHIp winds, the exchange ranges from -2.9 to 0.6 mmol m$^{-2}$ d$^{-1}$ with an average of -1.6 mmol m$^{-2}$ d$^{-1}$; using the SATp winds, the annual uptake is larger, being between -3.9 and 0.3 mmol m$^{-2}$ d$^{-1}$ with an average for 2000–2011 of -2.1 mmol m$^{-2}$ d$^{-1}$. The trend is the same for both products, with a decrease in the flux and an increase in the absorption of pCO$_2$ from the atmosphere.The average difference between the wind from satellite and the wind from SMHI give a value of 0.98 m s$^{-2}$ and have an influence of 0.34 mmol m$^{-2}$ d$^{-1}$ on the air-sea CO$_2$ flux. Our method to recompute the pCO$_2$ give a root mean square between 19.5 and 38.5 $\mu$atm which depend of the basin, this has an effect on the air-sea CO$_2$ flux of -1.2 mmol m$^{-2}$ d$^{-1}$.

The two methods to compute the air-sea CO$_2$ flux have been used, one from (Wanninkhof et al., 2009) where the results are described above, the second from Rutgersson et al. (2009). The second one used the water-side convection from a model Norman (2013). The mean difference between the two products are 1.2 mmol m$^{-2}$ d$^{-1}$. The higher difference is observed in 1999 (3.2 mmol m$^{-2}$ d$^{-1}$) and in 2006 (2.6 mmol m$^{-2}$ d$^{-1}$). The difference from the coefficient exchange is 0.088. At seasonal scale the difference on the two methods are higher in spring and summer (April to August) range between 4 mmol m$^{-2}$ d$^{-1}$ in April and 10 mmol m$^{-2}$ d$^{-1}$) in June. In winter, the difference is between 0.2 and 2.0 mmol m$^{-2}$ d$^{-1}$.

To conclude, the pCO$_2$ incertitude give a high variability in the air-sea CO$_2$ flux, the wind product influence the value more than the variability, and the difference is quite similar in all the time serie. The method influence the variability and it does not influence all the time serie in the same way.

### 3.2.3    Air–sea CO$_2$ flux climatology

The climatology of the flux displays high seasonal and spatial variability, ranging from –13. to 10 mmol m$^{-2}$ d$^{-1}$. On average, from 1998 to 2011, the entire Baltic Sea acts as a source of 1.2 mmol m$^{-2}$ d$^{-1}$. The result are different if we used the method from Rutgersson et al. (2009) which give 1.4 mmol m$^{-2}$ y$^{-1}$ and give a sink if we used the SATp winds -1.5 mmol m$^{-2}$ y$^{-1}$ (Fig. 10). The values observed are in agreement with those from other studies, indicating that the Baltic Sea can be a small source on average or a small sink of CO$_2$. Most previous research results concerning the carbon budget cover shorter periods, indicating a range between –1.16 and 2.9 mol m$^{-2}$ y$^{-1}$)(e.g. Wesslander et al., 2010; Kulinski and Pempkowiak, 2012), though the maximum values reported in these studies are all found in the same one or two years Algesten et al. (2006). Half of the studies demonstrate that Baltic Sea or certain basins of it act as sources, while the others demonstrate that it acts as a sink for the atmosphere (Norman et al., 2013a).

### 4    Discussion and Conclusions

Canadell (2003) explain that it is really challenging to estimate precisely the variation of the pCO$_2$ in marginal seas. This is due to several aspects but mainly due to temporal and spatial sparsity of measurements. Remote sensing using applicable

algorithms could certainly be an important approach, complementing ship-board observations as well as in situ buoy and wind-tower measurements. Using our method, we present the first estimated $CO_2$ flux climatology based on remote sensing for the Baltic Sea. This gives an estimated annual mean air–sea $CO_2$ flux of $1.2 \pm 0.8$ mmol m$^{-2}$ d$^{-1}$ and a seasonal variability of between –13. to 10 mmol m$^{-2}$ d$^{-1}$. The interannual variability is one order of magnitude lower, being between 0.01 and 3.19

5    mmol m$^{-2}$ d$^{-1}$. Several studies have estimated the air–sea $CO_2$ fluxes in the Baltic Sea over the last decade; most of these examine specific regions, but only a few treat the entire Baltic Sea. Kulinski and Pempkowiak (2012) demonstrate that the Baltic Sea was a source of $CO_2$ for the atmosphere between 2002 and 2008, but they use data from several time periods and sources. Using a biogeochemical model covering the 1960–2009 period, Norman et al. (2013b) suggest that the entire Baltic Sea acts as a net sink of between –0.22 and –0.17 mol m$^{-2}$ yr$^{-1}$, in agreement with our value of –0.6 mol m$^{-2}$ yr$^{-1}$.

In the Gulf of Findland, we found the lowest source of $CO_2$ from the atmosphere (0.2 mol m$^{-2}$ yr$^{-1}$), which ranges between —0.3 to 0.9 mol m$^{-2}$ yr$^{-1}$. These lowest value are observed in 2005 and 2007 to 2009: during this period it is actually a sink for the atmosphere. The gulf of Bothnia is a sink in 2009 in our study but this value decreases from 1998 to 2009. This flux has a value of 0.5 mmol m$^{-2}$ yr$^{-1}$ in 2002, lower than the value of 2.9 mol m$^{-2}$ yr$^{-1}$ from Algesten et al. (Algesten et al., 2006). This estimation is based on a few days of measurements from a few stations in the Gulf of Bothnia. Our results indicating a

small source are in agreement with those of the study demonstrating a larger sink in the Bothnian Sea (–0.73 mol m$^{-2}$ yr$^{-1}$) and a smaller source in Bothnian Bay (0.14 mol m$^{-2}$ yr$^{-1}$) between 1999 and 2009; this finding could explain why the entire Gulf of Bothnia region acts as a small sink or a small source on average.

In the Central Basin, Schneider et al. (2014) demonstrate that in four selected years (i.e. 2003,2004, 2009, and 2010), the surface water acts as a sink for the atmosphere, as found in our study, the value of the uptake rates ranging between –0.04 and

20    –0.3 mol m$^{-2}$ yr$^{-1}$. One study explain that the rates is the one which explain the enhance carbon in the sediments (Schneider et al., 2014). Our study of 2005, 2008, and 2009 finds an uptake value between –0.9 and –1.0 mol m$^{-2}$ yr$^{-1}$, slightly higher than that reported Schneider et al. (2014), who use boat-line data. This could be because of the spatial resolution of our product, which includes the entire Central Basin. Our mean value for the Central Basin indicates that it is a sink for the atmosphere. This is in contrast to the findings of Wesslander et al. (2010), who demonstrate that, for a slightly different period (i.e. 1994 to

2008), the Central Basin acts as a source for the atmosphere of 1.64 mol m$^{-2}$ yr$^{-1}$. As we explain in the Parard et al. (2014), the pCO$_2$ data set obtain do not reproduce the spring/summer bloom in the Eastern Gotland Sea described in (Schneider et al., 2015). The data used for the computation contain the VOS ship line but we made monthly average so we missed some higher frequency processes. In the study, they explain that the spring bloom take place around February 12 and March 21 (5 weeks), so the average must smooth the variability due to the bloom. In order to improve the pCO$_2$ data set, it will be better to used the

daily data in order to better reproduce such processes.

To conclude, in first approximation used remote sensing data and in-situ pCO$_2$ data to compute the FCO$_2$ gives good spatial and temporal resolutions compared with those of measurements from ships or wind-towers. The satellite data give information on pCO$_2$ variability and on FCO$_2$. The first estimates of Baltic Sea air–sea exchange based on remote-sensing products display reasonably good agreement with previous estimates and indicate a negative trend, with annual uptake changing from 0.6 to

35    –2.8 mol m$^{-2}$ yr$^{-1}$) over the 1998–2007 period. After 2007, the decrease is smaller and the flux remains quite stable at around

–2.8 mol m$^{-2}$ yr$^{-1}$). The air-sea $CO_2$ flux product depends on the wind product and on the $pCO_2$ product but also on the water convection. For winds, the higher-resolution product gives larger flux amplitudes, and for $pCO_2$, chlorophyll and CDOM are essential inputs.

The air–sea $CO_2$ flux is sensitive to different parameters as wind product in the Baltic Sea and the northern Baltic Sea. In the
5 Gulf of Bothnia, the wind plays affect the inter-annual variation in air–sea $CO_2$ flux which is higher than in the other basins. On average, the Central Basin near the South Basin is the region with the highest uptake of $CO_2$. The coastal region has a slightly higher uptake than does the open-sea region.

Several parameters are useful to improve our product as more in-situ data to constrains more our computation, but also used other parameters such salinity which has a strong variability in the Baltic Sea and a higher frequency in order to better represent
10 the different processes to better estimate the air-sea $CO_2$ flux.

*Acknowledgements.* We thank Dr. Tiit Kutser and Dr. Melissa Chierici for their help. We would also like to show our gratitude to the Prof. Sylvie Thiria for sharing their pearls of wisdom with us during the course of this research. This research was supported by Swedish National Space Board (grant no. 120/11:3).

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

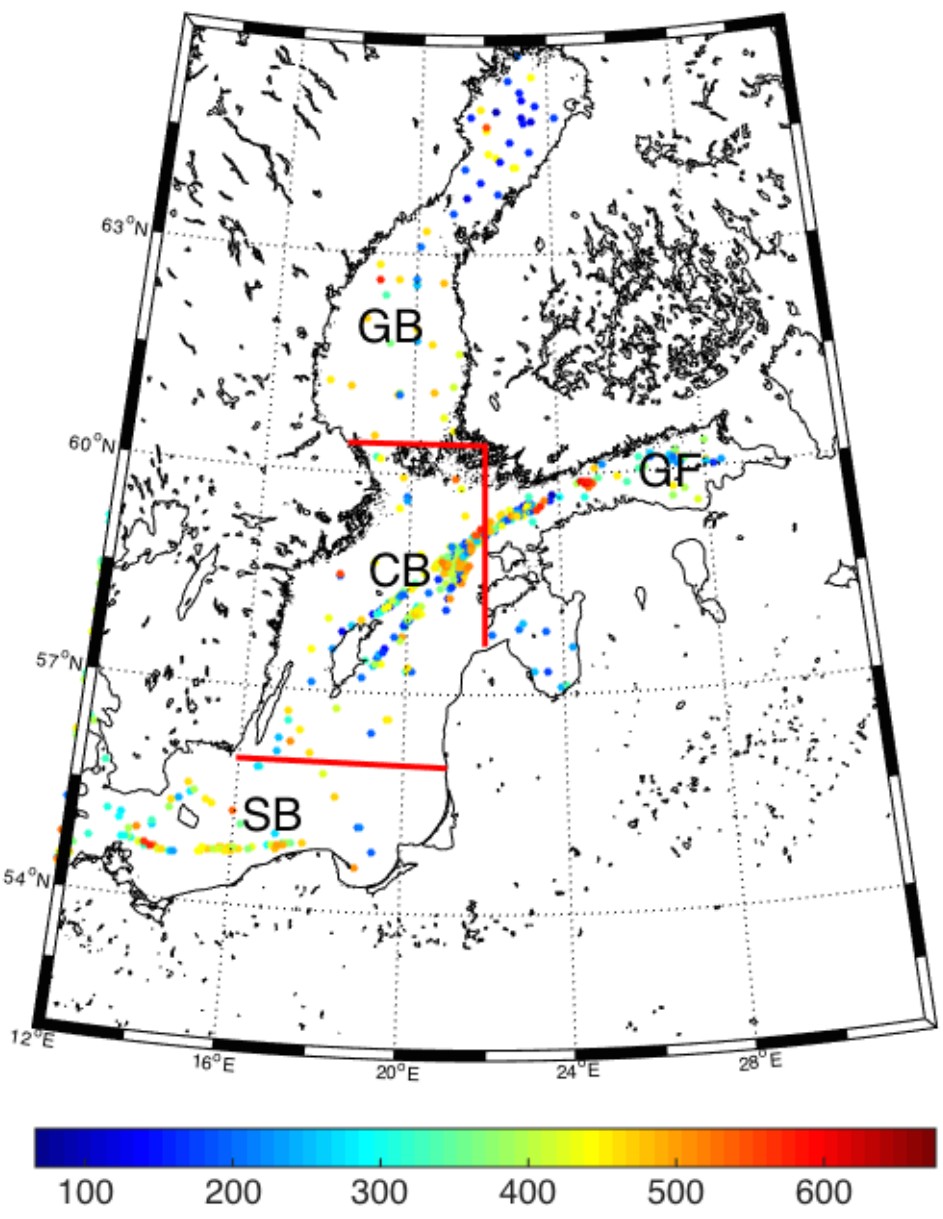

**Figure 1.** Data available for the Baltic Sea, 1998–2011. The red lines indicate the division into the Central Basin (CB), Gulf of Finland (GF), Gulf of Bothnia (GB), and South Basin (SB).

**Table 1.** RMS, bias, and correlation coefficients for in situ data from SMHI, Östergarnsholm wind-tower, and satellite products.

| Tower | SMHIp | | |
|---|---|---|---|
| | Bias | RMS | R |
| TOTAL | 0.67 | 2.49 | 0.84 |
| ÖSTERGARNSHOLM | 2.42 | 3.15 | 0.74 |
| FALSTERBO | 1.70 | 2.27 | 0.86 |
| HELSINGBORG | -0.88 | 1.65 | 0.85 |
| HANÖ | 3.64 | 4.07 | 0.88 |
| ÖLAND SÖDRA | 0.62 | 1.70 | 0.86 |
| HOBURG | -1.05 | 1.91 | 0.88 |
| NIDINGEN A | 3.68 | 4.17 | 0.85 |
| VINGA | 3.33 | 3.84 | 0.88 |
| ÖLAND NORRA | -0.29 | 1.52 | 0.87 |
| VISBY | -1.88 | 2.56 | 0.87 |
| MASESKAR | 3.82 | 4.29 | 0.91 |
| NORDKOSTER | 2.87 | 3.30 | 0.88 |
| HARSTENA | -0.33 | 1.45 | 0.86 |
| LANDSORT | 1.73 | 2.41 | 0.83 |
| GOTSKA | -1.60 | 2.20 | 0.91 |
| SVENSKA HÖGARNA | 1.57 | 2.31 | 0.8 |
| ÖRSKÄR | 1.07 | 2.02 | 0.86 |
| KUGGÖREN | -0.52 | 1.90 | 0.79 |
| BRÄMÖN | 0.29 | 1.86 | 0.78 |
| SKAGSUDDE | -0.37 | 1.78 | 0.79 |
| HOLMOGADD | -0.60 | 1.85 | 0.82 |
| HOLMÖN | -0.75 | 2.13 | 0.78 |
| BJURÖKLUBB | 0.13 | 2.16 | 0.75 |
| LULEÅAIRPORT | -2.32 | 3.17 | 0.68 |

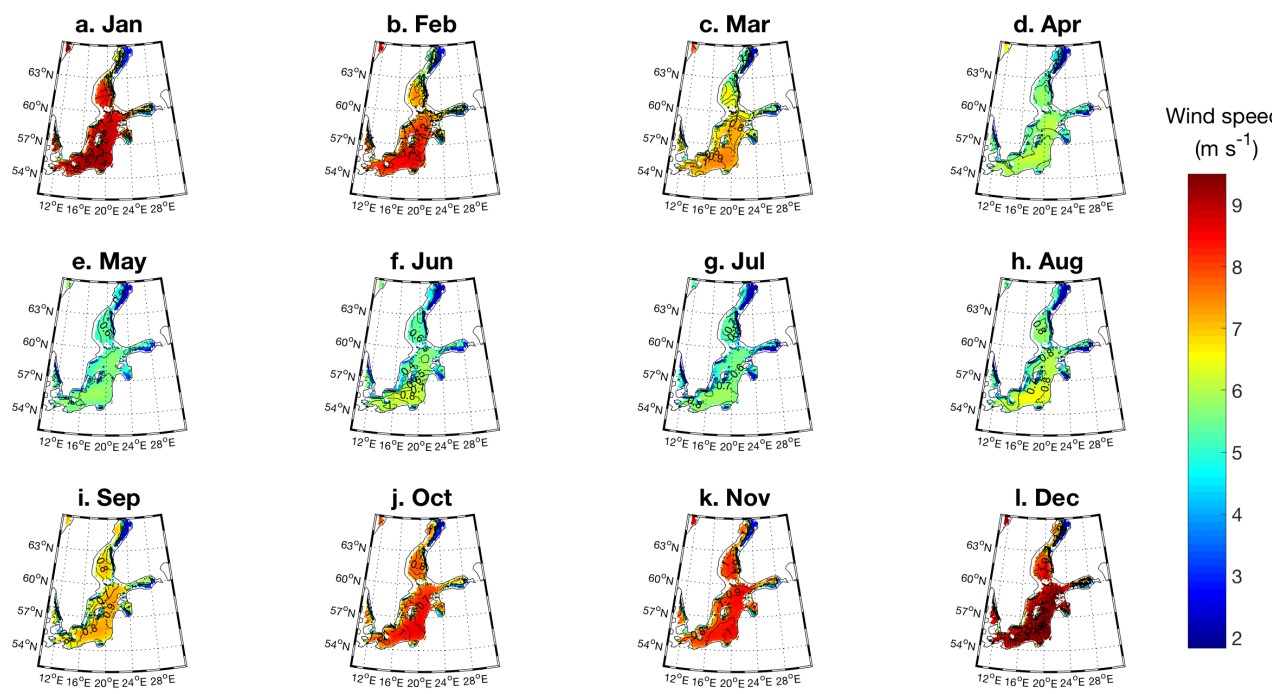

**Figure 2.** Monthly mean wind speed (indicated by colour bar) and annual variability (indicated by contours).

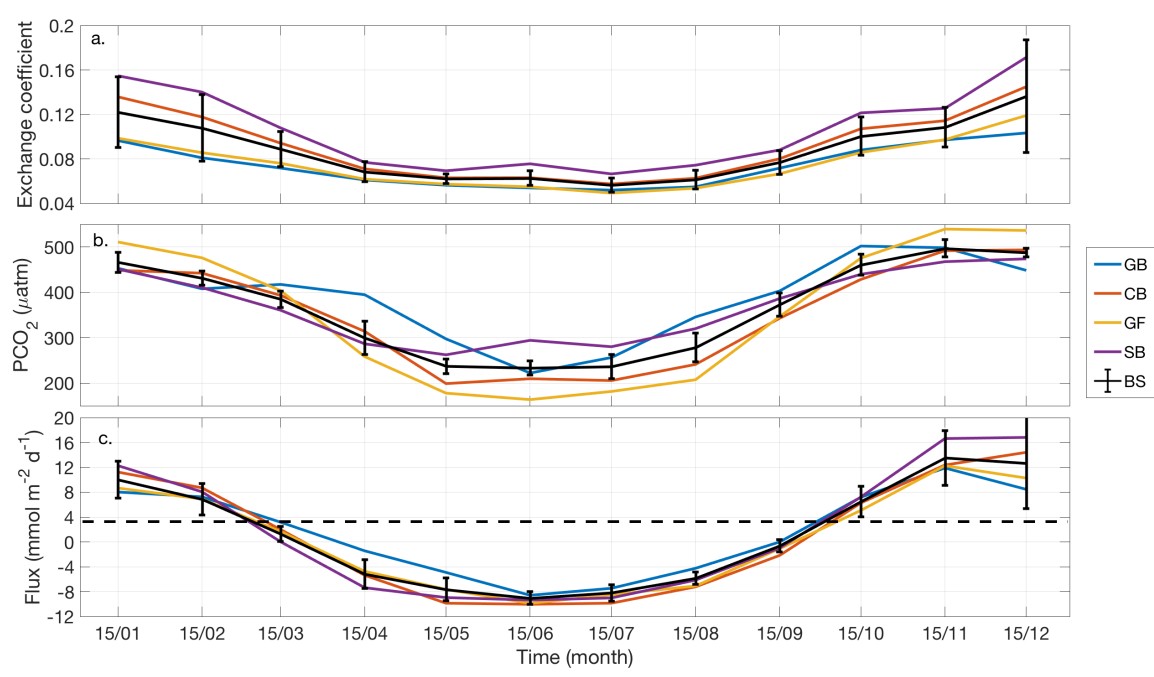

**Figure 3.** Evolution annual of the a.) Transfert velocity based on Wanninkhof et al. (2009). b.) $PCO_2$ and c.) air-sea $CO_2$ flux based on the SMHIp wind product for each bassin.

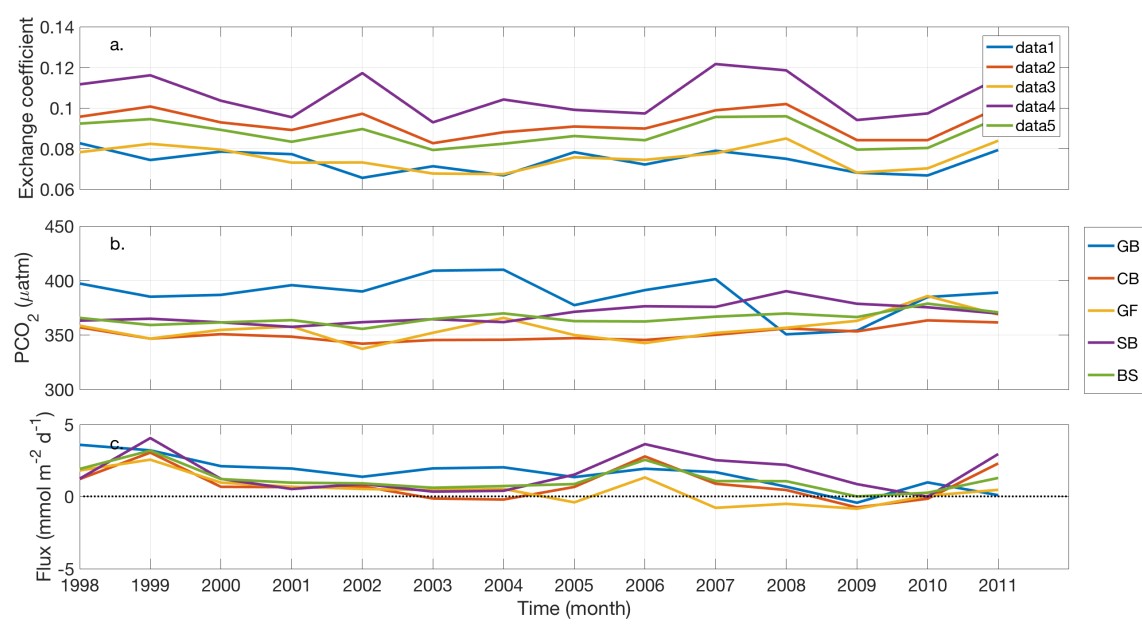

**Figure 4.** Evolution annual of the a.) Transfert velocity based on Wanninkhof et al. (2009). b.) $PCO_2$ and c.) air-sea $CO_2$ flux based on the SMHIp wind product for each bassin.The abreviation correspond to the basin GB : Gulf of Bothnia, CB : Central Basin; GF Gulf of Findland; SB : South Basin and BS: Baltic Sea.

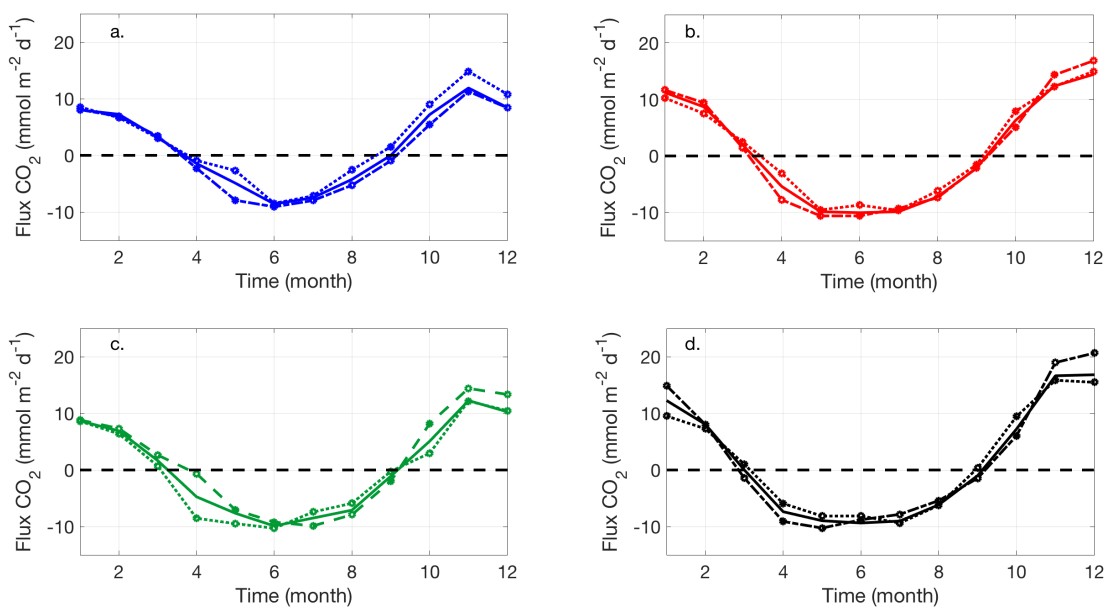

**Figure 5.** Seasonal cycle of air-sea CO2 flux for a) Gulf of Bothnia, b) Central Baltic c) Gulf of Finland and d) Southern Baltic. Solid lines represent the average for the full period (1998 to 2011), dotted lines with markers are for the first 5 years (1998-2002) and dashed lines are for the last five years (2007 to 2011).

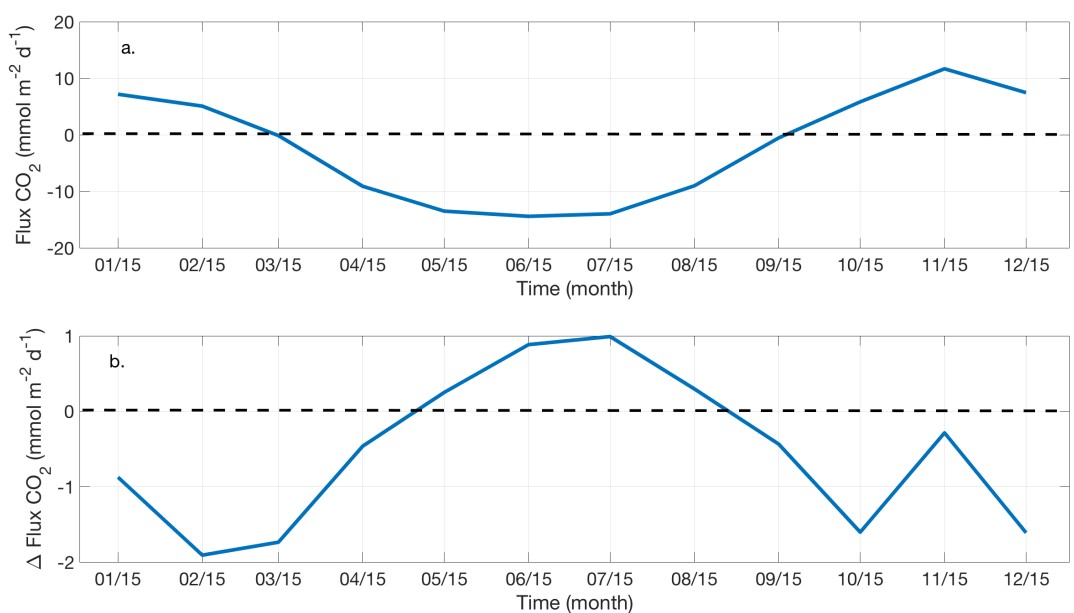

**Figure 6.** Average, 1998–2011, a) of the air–sea $CO_2$ flux and b) of the difference between the coastal region and open sea.

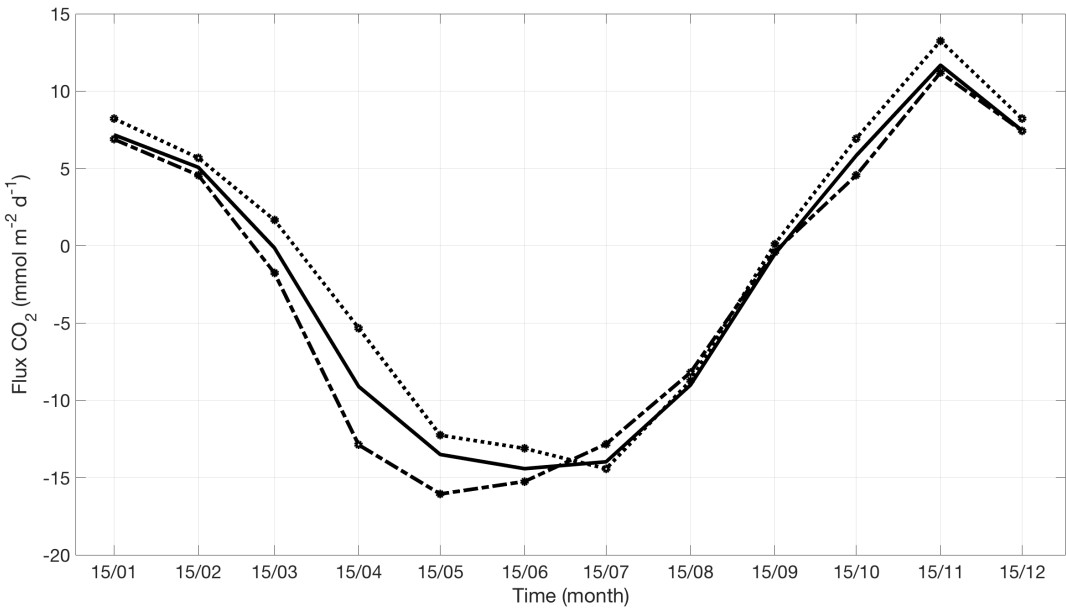

**Figure 7.** Seasonal cycle of air-sea $CO_2$ flux for Baltic Sea. Solid line represent the average for the pull period (1998–2011), dotted linewith marker is for the first 5 years (1998-2002) and dashed line is for the last fiver year (2007 to 2011).

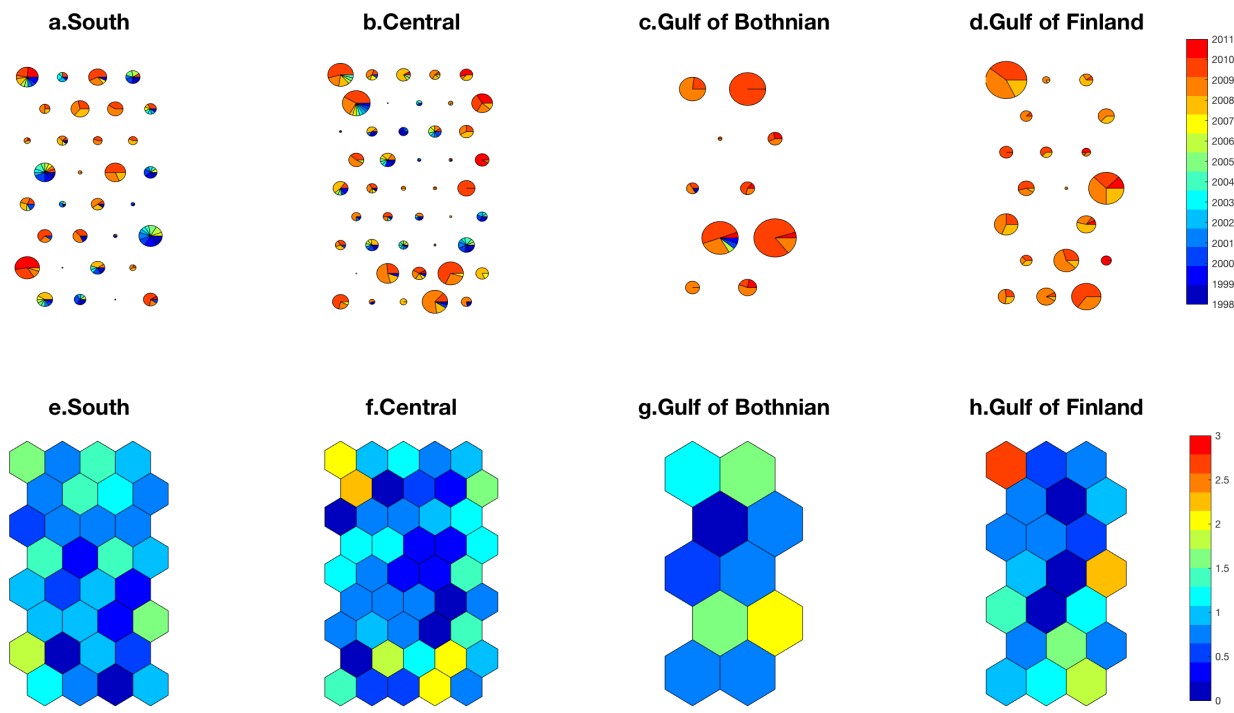

**Figure 8.** a.,b.c. and d. are the distribution of the years of each data in each class for each basin SOM e.,f.,g. and h. are the percentage of the total data present in each class of the different basins' SOM. The size of the circles in the top figures is also representative of the percentage of the total data present in each class of the different basins' SOM.

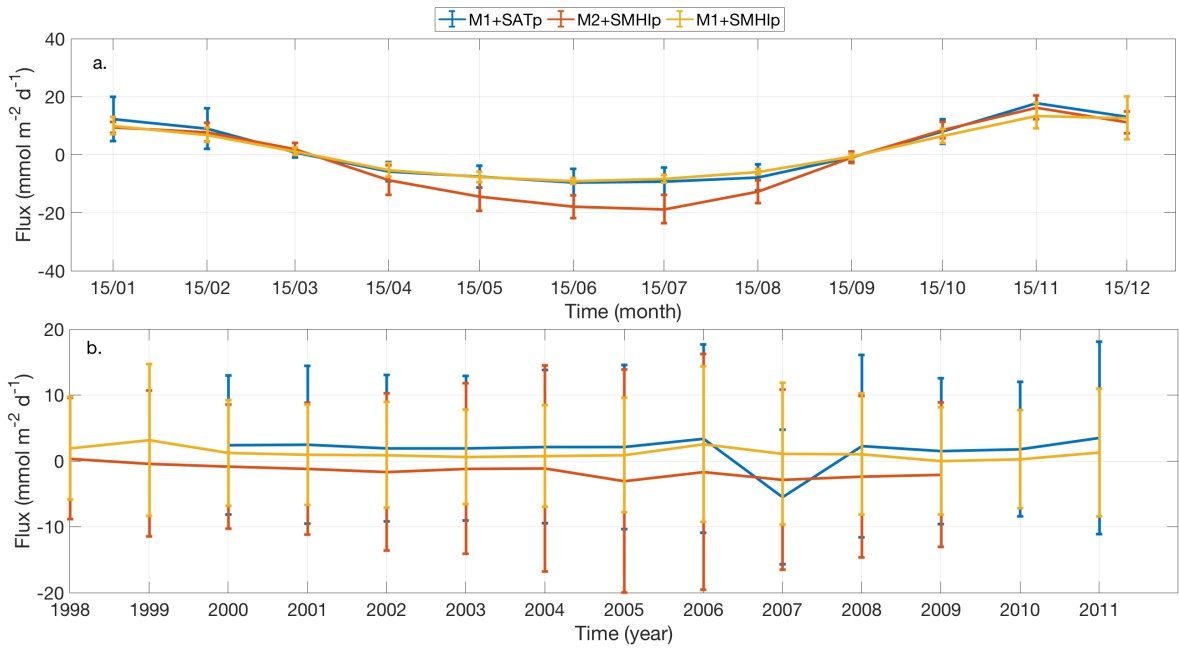

**Figure 9.** The air-sea $CO_2$ flux estimate evolution with method 1 and the SATp product (Blue); method 2 and the SMHIp product (Red); method 1 and the SMHIp product (Yellow). a. for a year b. in average for all the year.

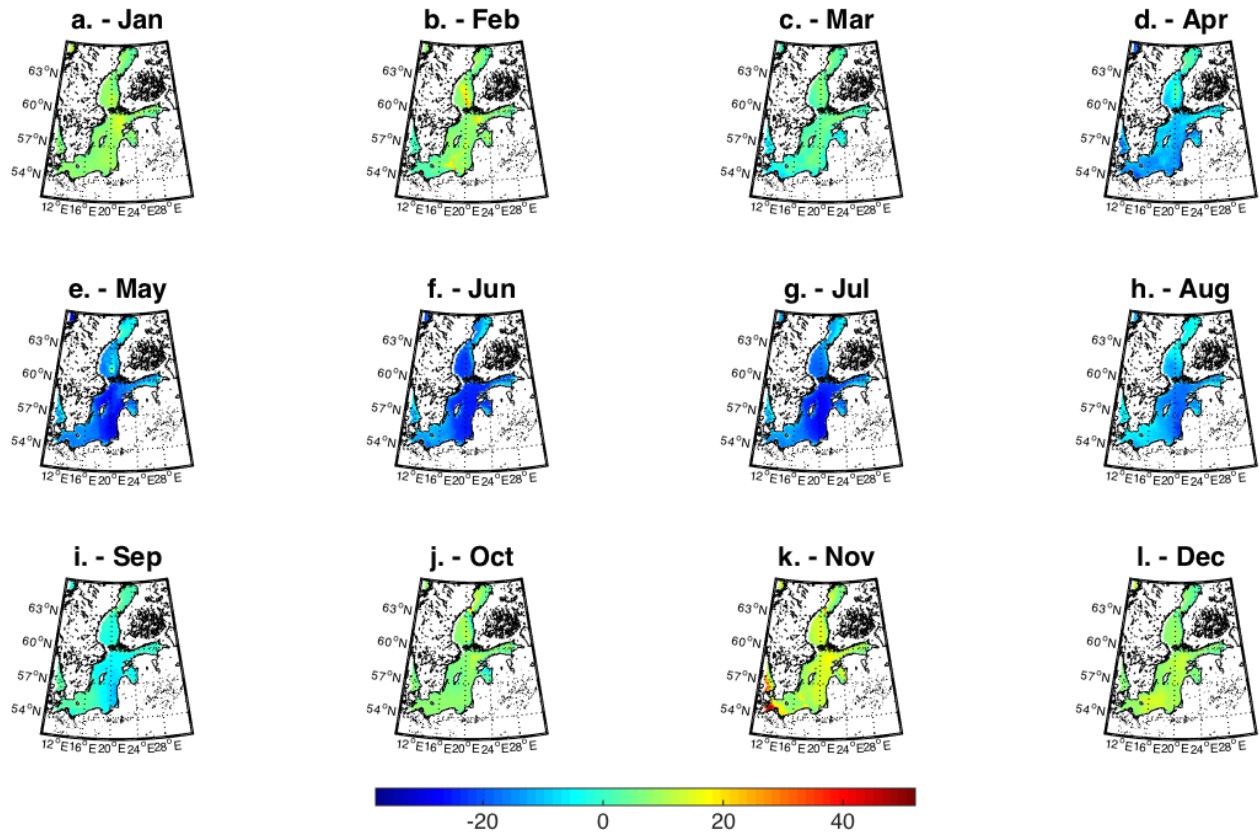

**Figure 10.** Temporal evolution of the air–sea CO$_2$ flux between 1998 and 2011 based on SMHIp data.

Interactive comment on "The Potential of using Remote Sensing data to estimate Air–Sea CO2 exchange in the Baltic Sea" by Gaëlle Parard et al.

Anonymous Referee #2

The study by Parard et al. focuses on the very important and interesting aspect of the present-day oceanography, namely on the role of coastal and marginal seas in the global carbon cycle. There is an ongoing debate in the scientific literature if these regions act as sink or source of CO2. Parard et el. propose to use for the studies on CO2 fluxes in coastal regions remote sensing tools. In the revised manuscript, they present results from the Baltic Sea. The worldwide context (though poorly presented in the paper) and importance of the problem raised by the authors places, in my opinion, the manuscript within the scope of interests of Earth System Dynamics. However, the manuscript should be first improved in several aspects mentioned below and thus requires further revision.
General comments:

1). The goal of the presented manuscript is ambiguous. It is unclear what is the novelty in the presented research especially in the context of previous publications of the authors in the field. Please specify clearly what is the added value of the presented study.
The goal of the paper is to present from 2005-2011 the air-sea CO2 flux variability computed with $pCO_2$ estimated from satellite data in order to study the variability at a seasonal and interannual scale. We will rewrite this part in order to clarify the message (p1-2 introduction)

2). The importance of the study could be better presented in the worldwide context of carbon cycling and role of the coastal and marginal seas.
We will rewrite this part to better present the worldwide context (p1-2 introduction)

3). The manuscript should contain better review on the pCO2 fields and CO2 fluxes reported for the Baltic Sea in the recent years. There were several papers published on that recently. Important contribution to that issues are also regular measurements of pCO2 made on the VOS line operated by IOW between Germany and Finland. This comment refers to the entire manuscript but especially to the introduction section where only the paper by Wesslander et al. (2010) is mentioned in that context.
The pCO2 from the VOS line were used to compute the pCO2 in our study like it is presented in the paper Parard et al., 2016. In order to improve the paper we will better present a review of the pCO2 field and in particular the air-sea CO2 flux (3.2.1 and discussion and conclusion)

4). The methods used in the study are not well described and documented. It is relatively clear how the winds data were established. However it is unclear how the remote sensing data are transferred into pCO2. I am aware of the ongoing debate on the obstacles with the application of remote sensing in the Baltic Sea. Since I am not an expert on remote sensing I do not want to judge on that. However, at least the limitations of the remote sensing methods should be discussed in the manuscript in the context of pCO2 calculations.

The method is fully described in the method paper Parard et al.,2016, and the plagiarism software forced us to remove everything in that section. So in this paper we wanted to focus on the air-sea CO2 flux variability. We will try to rewrite the method part in a way that the plagiarism software finds non plagiarizing but it has been quite difficult in our past 3 tries (Data and method 2.1 p3-4)

5). The CO2 flux across the air/sea interface is a function of the wind speed and pCO2 difference between seawater and the atmosphere. Both these parameters are critical for accurate CO2 flux estimations. It would be meaningful to demonstrate that the pCO2 fields obtained from the remote sensing data are correct. This could be done by comparison with the available pCO2 measurements.
7). How the accuracy in the determination of pCO2 fields influence the calculated CO2 fluxes? The latter, as it appears from Fig.8, are burdened with a relatively high uncertainty.

The comparison for pCO2 is already done in the paper parard et al.2015 and 2016 , we add in the part titled "uncertainty analysis" a discussion on the influence of the pCO2. (3.2.2 uncertainty analysis)

6). Experimental data suggest that there are two minima in seasonality of pCO2 in the Eastern Gotland Basin, which are related to the spring bloom and mid-summer N2 fixation. Why this is not seen in the modelled pCO2 (Fig. 2)? Please comment on that.
The monthly data do not allow us to catch the process we add a discussion part 4 (p10)

8). Presenting the results as annual means is not very informative. Fig. 3b gives the impression that seawater is permanently undersaturated with CO2 (seawater pCO2 lower from the atmospheric one). This is misleading.
We change this part but we keep the annual variability in agreement with some other comments from other reviewer.

9). The entire manuscript requires careful editing. Now it contains number of technical defects. As a part of this work English could be also improved. However I leave this as a suggestion only as English is not my mother tongue.
We try to improve the english

Minor comments:

10). It would be meaningful to add a map of the Baltic Sea showing the places mentioned in the manuscript.
We add a map with the in situ data used (Figure 1)

11). Page 2, line 25. Not the best choice of references – paper by Omstedt et al. 2009 does not refer to the global scale
We change that p2 l27

12). Please add how big the river runoff is (page 2, line 32) 13). Page 3, line12. Mixed layer depth is not always on 60m.
We change the two, (p2 l35 p3 l30).

14). Section 3.2.1. The discussion on seasonal and annual means are mixed up in the text. This causes that it is difficult for the reader to follow the text.
We rewrite this part in order to make it clearer.

15). Page 6, line22. I think it should be Fig. 3. 16). Page 6, line 30. Fig. 3 does not show seasonality
We will correct that (p7 l18 and 10)

17). Page 6, line 30. Outgassing can happened only when seawater pCO2 is higher from the atmospheric one. It is impossible in summer in open sea.
We correct that (p7 l18)

18). Page 7, line 7. Please name these different satellite products.
We rewrite this sentence (p8 l15)

19). Page 7, line 12. "flux from the coastal region" – this suggests flux in only one direction – please rephrase.
We rewrite this part (p8 l20)

20). Page 7, line34. What data this refers to? Fig. 3 shows data for GF also for the period before 2008.
It was a mistake it was for the GB (p8 l25)

21). Page 8, line 2. Should be these
We change that (p8 l30)

22). Page 8, line 15. Wrong unit of the wind speed
No it is the impact of the wind on the air sea CO2 flux.

23). Page 8, line 16. "in function of the basin" – unclear.
We change this sentence (P9 l9)

24). Page 8, line 26. Please rephrase
We detail this part (3.2.3)

25). Page 9, line 1. Over or in the marginal seas
We correct that (P9 l31)

26). Page 9, line 7. Please reduce the number of figures after comma.
We change that (p10 l5)

27). Page 9, line 11. Please correct citation.
We change that (p10 l9)

28). The abbreviations of the different water basins (GB, CB, GF, SB, BS) should be explained when first time used in the paper 29). Fig. 3a, name data 1, data 2 etc.
We change that

Interactive comment on "The Potential of using Remote Sensing data to estimate Air–Sea CO2 exchange in the Baltic Sea" by Gaëlle Parard et al.

Anonymous Referee #1

The manuscript by Parard et al. is on a very interesting topic – the role of coastal waters (the whole Baltic Sea belongs to them) in the carbon cycle and using remote sensing in determining the role. It is obvious that most of the carbon processing is taking place in coastal waters (where the amount of carbon in different forms is the highest). On the other hand this is also the area where remote sensing has the biggest problems due to optical complexity of the waters, atmospheric correction issues (the assumptions used in ocean remote sensing are not valid in coastal waters) as well as the adjacency effects present close to the shores. The Baltic Sea is a particularly complicated study object due to it's low reflectance (high concentration of CDOM) and low sun angles during most of the year. Therefore, the remote sensing part is the weakest link in this study.

First of all the remote sensing methodology part is not well described in the manuscript in order to be able to understand the potential errors of the methodology used. It is understandable from the Authors point of view that if they have published already two similar studies where the methodology was described in more detail then they kind of assume that the methodology works. Moreover, plagiarism detection software picks it up very easily if the methods description is repeated in several papers. On the other hand each manuscript has to be self-consistent. It is important from the readers perspective to understand what has been done without digging into databases, download- ing relevant papers, and learning what kind of methodology was used to produce the results.

We appreciate the closeness with which the reviewer examined our work. As he notes, the pCO2 used in this paper to compute the air-sea CO2 flux is the one described in the paper Parard et al, 2016. We initially developed the methodology more but the plagiarism software did not allow us to submit until we had removed too much of that part to keep any of it (Part data and method) .

Digging into the databases and reading the previous papers by Parard et al on the same topic revealed that there are serious issues with the remote sensing products used. It is said for Chl-a that SeaWiFS and MODIS monthly means were used. It is not said which algorithm was used as the reference added there is about AVHRR not these two satellites. One can assume that OC4 type blue-green band ratio was used as this kind of algorithms are standard for these sensors. It has been known for many decades that blue-green ratios do not work in coastal and inland waters, especially in CDOM-rich waters like the Baltic Sea. This has been demonstrated by Darecki and Stramski 2004, Reinart and Kutser 2006, Ligi et al. 2017 and many others. The latest study used mainly modelled data. Meaning perfect reflectance values in that sense that there were no atmospheric correction errors. Still, the latest version of OC4 tuned for the Balti Sea gave correlations that were close to zero. The Copernicus Marine Environment Monitoring Service (CMEMS) validated the MODIS Chl-a algorithm for the Baltic Sea and got correlations r2=0.2. The new CMEMS product is based on a neural network approach but still their validation results show the correlation with in situ data is

r2=0.2. This means that one of the main input products used by the Authors has very little to do with actual chlorophyll-a values in the Baltic Sea.

The second product used by the Authors is CDOM. Again, an open ocean algorithm (Morel and Gentili 2009) was used to create the CDOM product. It is known not to work in coastal waters, especially in waters with high-CDOM like the Baltic Sea. There are several papers by Kowalczuk et al., Kratzer et al. and others where CDOM algorithms that produce realistic CDOM estimates for the Baltic Sea have been proposed. Proba- bly the CMEMS previous version used the same algorithm as the Authors in their study, but CMEMS did not provide CDOM validation result for the Baltic Sea. Most likely, be- cause the correlation with Baltic Sea CDOM was far lower than for Chl-a. The new (neural network based) CMEMS CDOM product has not been validated at all. Thus, the Authors used another remote sensing product that does not work in the Baltic Sea (and I cannot provide a recommendation where to download a reasonable product).

The third remote sensing product used is primary production. First of all, it is a Chl-a based calculation and the Chl-a product used by the Authors has very little to do with the actual chlorophyll in the Baltic Sea, as was mentioned above. The NPP model used is also for oceanic not Baltic Sea type waters. I am not sure how much does this affect the results, but it is sure that using a model not designed for the Baltic Sea with input product that is useless for the Baltic Sea should not provide very realistic results. Baltic Sea specific primary production models were published also more than 20 years ago (Wozniak et al. 1995 and several other papers by the same authors). So, better and more relevant NPP models exist. Without proper validation I do not trust the currently used NPP model.

As a remote sensing scientist I would like to see the remote sensing methods used in as many applications as possible. On the other hand, it hurts to see that people use different remote sensing product in their studies without checking are these products realistic or not. Huge amount of work has been done, but maybe only the spatial patterns found in the study have some connections with the real pCO2 fields in the Baltic Sea. I definitely do not trust in any numbers currently shown in the manuscript as at least three of the input products that cannot be used in the Baltic Sea were used in this study.

We are agreeing about the inconsistence of the numerical values of our input database but in our case, we needed the physical variability in space in time to be coherent more than the exact value. The neural method is capable of learning to estimate the values wanted from this underlying physics independently on the correctness of their numerical value, provided the sources represent the underlying dynamics in a coherent way and we take the same inputs as the ones the algorithm was trained with.

Furthermore, for the inference of pCO2, the correlation of each input element to it is taken into account, so in this case of the coastal region, the impact of Chl-A will be lessened. It is going to play a role in the central part of the basin, however.

Unfortunately, the concerns raised by the reviewer, who is clearly an expert in remote sensing data, cannot be discussed without a better explanation of the details of the previous papers, and it seems the critiques are mostly on our previous work. Furthermore, even if the quality of the remote sensing data is poor, it does not mean that our results are not coherent. We will however add a part in the discussion to say that the values obtain

are dependant of the hypothesis that the observations representing at least the dynamic physic of the basin.