# Peer review of "The Potential of using Remote Sensing data to estimate Air-Sea CO2 exchange in the Baltic Sea."

_Earth System Dynamics, 2017_

## Referee Comment (RC1) · Anonymous Referee #1 · 19 May 2017

[referee-annotated manuscript omitted]

---

## Referee Comment (RC2) · Anonymous Referee #2 · 21 Jun 2017

The study by Parard et al. focuses on the very important and interesting aspect of the present day oceanography, namely on the role of coastal and marginal seas in the global carbon cycle. There is an ongoing debate in the scientific literature if these regions act as sink or source of CO2. Parard et el. propose to use for the studies on CO2 fluxes in coastal regions remote sensing tools. In the revised manuscript they present results from the Baltic Sea. The worldwide context (though poorly presented in the paper) and importance of the problem raised by the authors places, in my opinion, the manuscript within the scope of interests of Earth System Dynamics. However, the manuscript should be first improved in several aspects mentioned below and thus requires further revision.

General comments: 1). The goal of the presented manuscript is ambiguous. It is unclear what is the novelty in the presented research especially in the context of previous publications of the authors in the field. Please specify clearly what is the added value of the presented study. 2). The importance of the study could be better presented in the worldwide context of carbon cycling and role of the coastal and marginal seas. 3). The manuscript should contain better review on the pCO2 fields and CO2 fluxes reported for the Baltic Sea in the recent years. There were several papers published on that recently. Important contribution to that issues are also regular measurements of pCO2 made on the VOS line operated by IOW between Germany and Finland. This comment refers to the entire manuscript but especially to the introduction section where only the paper by Wesslander et al. (2010) is mentioned in that context. 4). The methods used in the study are not well described and documented. It is relatively clear how the winds data were established. However it is unclear how the remote sensing data are transferred into pCO2. I am aware of the ongoing debate on the obstacles with the application of remote sensing in the Baltic Sea. Since I am not an expert on remote sensing I do not want to judge on that. However, at least the limitations of the remote sensing methods should be discussed in the manuscript in the context of pCO2 calculations. 5). The CO2 flux across the air/sea interface is a function of the wind speed and pCO2 difference between seawater and the atmosphere. Both these parameters are critical for accurate CO2 flux estimations. It would be meaningful to demonstrate that the pCO2 fields obtained from the remote sensing data are correct. This could be done by comparison with the available pCO2 measurements. 6). Experimental data suggest that there are two minima in seasonality of pCO2 in the Eastern Gotland Basin, which are related to the spring bloom and mid-summer N2 fixation. Why this is not seen in the modelled pCO2 (Fig. 2)? Please comment on that. 7). How the accuracy in the determination of pCO2 fields influence the calculated CO2 fluxes? The latter, as it appears from Fig.8, are burdened with a relatively high uncertainty. 8). Presenting the results as annual means is not very informative. Fig. 3b gives the impression that seawater is permanently undersaturated with CO2 (seawater pCO2 lower from the atmospheric one). This is misleading. 9). The entire manuscript requires careful editing.

Now it contains number of technical defects. As a part of this work English could be also improved. However I leave this as a suggestion only as English is not my mother tongue.

Minor comments: 10). It would be meaningful to add a map of the Baltic Sea showing the places mentioned in the manuscript. 11). Page 2, line 25. Not the best choice of references – paper by Omstedt et al. 2009 does not refer to the global scale 12). Please add how big the river runoff is (page 2, line 32) 13). Page 3, line12. Mixed layer depth is not always on 60m. 14). Section 3.2.1. The discussion on seasonal and annual means are mixed up in the text. This causes that it is difficult for the reader to follow the text. 15). Page 6, line22. I think it should be Fig. 3. 16). Page 6, line 30. Fig. 3 does not show seasonality 17). Page 6, line 30. Outgassing can happened only when seawater pCO2 is higher from the atmospheric one. It is impossible in summer in open sea. 18). Page 7, line 7. Please name these different satellite products. 19). Page 7, line 12. "flux from the coastal region" – this suggests flux in only one direction – please rephrase. 20). Page 7, line34. What data this refers to? Fig. 3 shows data for GF also for the period before 2008. 21). Page 8, line 2. Should be these 22). Page 8, line 15. Wrong unit of the wind speed 23). Page 8, line 16. "in function of the basin" – unclear. 24). Page 8, line 26. Please rephrase 25). Page 9, line 1. Over or in the marginal seas 26). Page 9, line 7. Please reduce the number of figures after comma. 27). Page 9, line 11. Please correct citation. 28). The abbreviations of the different water basins (GB, CB, GF, SB, BS) should be explained when first time used in the paper 29). Fig. 3a, name data 1, data 2 etc.

---

## Author Comment (AC1) · 21 Jul 2017

[revised manuscript text omitted]

We agree about the inconsistence of the numerical values of our input database but in our case, we needed the physical variability in space in time to be coherent more than the exact numerical value. The neural method is capable of learning to estimate the values wanted from this underlying physics independently on the correctness of their numerical value, provided the sources represent the underlying dynamics in a coherent way and we take the same inputs as the ones the algorithm was trained with.

Furthermore, for the inference of pCO2, the correlation of each input element to it is taken into account, so in this case of the coastal region, the impact of Chl-A will be lessened. It is going to play a role in the central part of the basin, however.

Unfortunately, the concerns raised by the reviewer, who is clearly an expert in remote sensing data, cannot be discussed without a better explanation of the details of the previous papers, and it seems the critiques are mostly on our previous work. Furthermore, even if the quality of the remote sensing data is poor, it does not mean that our results are not coherent. We will however add a part in the discussion to clearly discussion the fact

that the values obtain are dependant of the hypothesis that the observations representing at least the dynamic physic of the basin and the problems of the accuracy of the numerical values of the input data.

---

## Author Comment (AC2) · 21 Jul 2017

The study by Parard et al. focuses on the very important and interesting aspect of the present-day oceanography, namely on the role of coastal and marginal seas in the global carbon cycle. There is an ongoing debate in the scientific literature if these regions act as sink or source of CO2. Parard et el. propose to use for the studies on CO2 fluxes in coastal regions remote sensing tools. In the revised manuscript, they present results from the Baltic Sea. The worldwide context (though poorly presented in the paper) and importance of the problem raised by the authors places, in my opinion, the manuscript within the scope of interests of Earth System Dynamics. However, the manuscript should be first improved in several aspects mentioned below and thus requires further revision.
General comments:

1). The goal of the presented manuscript is ambiguous. It is unclear what is the novelty in the presented research especially in the context of previous publications of the authors in the field. Please specify clearly what is the added value of the presented study.

The goal of the paper is to present from 2005-2011 the air-sea CO2 flux variability computed with $pCO_2$ estimated from satellite data in order to study the variability at a seasonal and interannual scale. We will rewrite this part in order to clarify the message. Previous studies from the group has not focused on the flux or the flux variability.

2). The importance of the study could be better presented in the worldwide context of carbon cycling and role of the coastal and marginal seas.
We will rewrite this part to better present the worldwide context.

3). The manuscript should contain better review on the pCO2 fields and CO2 fluxes reported for the Baltic Sea in the recent years. There were several papers published on that recently. Important contribution to that issues are also regular measurements of pCO2 made on the VOS line operated by IOW between Germany and Finland. This comment refers to the entire manuscript but especially to the introduction section where only the paper by Wesslander et al. (2010) is mentioned in that context.

The pCO2 from the VOS line were used to compute the pCO2 in our study like it is presented in the paper Parard et al., 2016. In order to improve the paper we will better present a review of the pCO2 field and in particular the air-sea CO2 flux. We agree that there are several studies (in particular from German and Polish groups) to include.

4). The methods used in the study are not well described and documented. It is relatively clear how the winds data were established. However it is unclear how the remote sensing data are transferred into pCO2. I am aware of the ongoing debate on the obstacles with the application of remote sensing in the Baltic Sea. Since I am not an expert on remote sensing I

do not want to judge on that. However, at least the limitations of the remote sensing methods should be discussed in the manuscript in the context of pCO2 calculations.

The method is fully described in the method paper Parard et al.,2016, and the plagiarism software forced us to remove everything in that section. So in this paper we wanted to focus on the air-sea CO2 flux variability. We will try to rewrite the method part in a way that the plagiarism software finds non plagiarizing.

5). The CO2 flux across the air/sea interface is a function of the wind speed and pCO2 difference between seawater and the atmosphere. Both these parameters are critical for accurate CO2 flux estimations. It would be meaningful to demonstrate that the pCO2 fields obtained from the remote sensing data are correct. This could be done by comparison with the available pCO2 measurements.

The part titled "uncertainty analysis" is where we develop all the discussion about the impact of the parameter on the air-sea CO2 flux. For the validation of the pCO2 field from the satellite data, it is already done in Parard et al,2016. To clarify the message we will add a discussion develop this part.

6). Experimental data suggest that there are two minima in seasonality of pCO2 in the Eastern Gotland Basin, which are related to the spring bloom and mid-summer N2 fixation. Why this is not seen in the modelled pCO2 (Fig. 2)? Please comment on that.

This is a good question and we need to develop this part, we used monthly mean data, it could be the reason that we missed this signal.

7). How the accuracy in the determination of pCO2 fields influence the calculated CO2 fluxes? The latter, as it appears from Fig.8, are burdened with a relatively high uncertainty.

As explain in 5 we need to develop this part.

8). Presenting the results as annual means is not very informative. Fig. 3b gives the impression that seawater is permanently undersaturated with CO2 (seawater pCO2 lower from the atmospheric one). This is misleading.

We will present the result differently in order to clarify the message.

9). The entire manuscript requires careful editing. Now it contains number of technical defects. As a part of this work English could be also improved. However I leave this as a suggestion only as English is not my mother tongue.

We will correct the manuscript.

Minor comments:

10). It would be meaningful to add a map of the Baltic Sea showing the places mentioned in the manuscript.

We will do that.

11). Page 2, line 25. Not the best choice of references – paper by Omstedt et al. 2009 does not refer to the global scale

We will change that

12). Please add how big the river runoff is (page 2, line 32) 13). Page 3, line12. Mixed layer depth is not always on 60m.

We will change that

14). Section 3.2.1. The discussion on seasonal and annual means are mixed up in the text. This causes that it is difficult for the reader to follow the text.

We will correct that

15). Page 6, line22. I think it should be Fig. 3. 16). Page 6, line 30. Fig. 3 does not show seasonality

We will correct that

17). Page 6, line 30. Outgassing can happened only when seawater pCO2 is higher from the atmospheric one. It is impossible in summer in open sea.

We will correct that

18). Page 7, line 7. Please name these different satellite products.

We will add the name

19). Page 7, line 12. "flux from the coastal region" – this suggests flux in only one direction – please rephrase.

We will correct that

20). Page 7, line34. What data this refers to? Fig. 3 shows data for GF also for the period before 2008.

We will rewrite this part

21). Page 8, line 2. Should be these

We will correct that

22). Page 8, line 15. Wrong unit of the wind speed

We will correct that

23). Page 8, line 16. "in function of the basin" – unclear.

We will correct that

24). Page 8, line 26. Please rephrase

We will rephrase

25). Page 9, line 1. Over or in the marginal seas

We will correct that

26). Page 9, line 7. Please reduce the number of figures after comma.

We will change that

27). Page 9, line 11. Please correct citation.

We will change that

28). The abbreviations of the different water basins (GB, CB, GF, SB, BS) should be explained when first time used in the paper 29). Fig. 3a, name data 1, data 2 etc.
We will change that

We will change that

---

## Editor Decision (ED1)

Mode: Similarity Report ▼

**paper text:**

[revised manuscript text omitted]

6    53 words / 1% - Internet from 17-Mar-2016 12:00AM
www.biogeosciences.net

7    48 words / 1% - Crossref
FERIAL LOUANCHI. "Modelled and observed sea surface fCO2 in the southern ocean: a comparative study", Tellus B, 4/1999

8    48 words / 1% - Crossref
Regional Climate Studies, 2015.

9    47 words / 1% - Crossref
Dong, Fang, Yangchun Li, Bin Wang, Wenyu Huang, Yanyan Shi, and Wenhao Dong. "Global Air–Sea CO2 Flux in 22 CMIP5 Models: Multiyear Mean and Interannual Variability*", Journal of Climate, 2016.

10    39 words / 1% - Crossref
Black, K.S.. "An autonomous benthic lander:", Continental Shelf Research, 200105/06

11    36 words / 1% - Crossref
Ocean-Atmosphere Interactions of Gases and Particles, 2014.

12    35 words / 1% - Crossref
Alberto V Borges. "Net ecosystem production and carbon dioxide fluxes in the Scheldt estuarine plume", BMC Ecology, 2008

13    30 words / 1% - Internet from 21-Dec-2015 12:00AM
oceanrep.geomar.de

14    29 words / < 1% match - Crossref
Alberto V. Borges, Cédric Morana, Steven Bouillon, Pierre Servais, Jean-Pierre Descy, François Darchambeau. "Carbon Cycling of Lake Kivu (East Africa): Net Autotrophy in the Epilimnion and Emission of CO2 to the Atmosphere Sustained by Geogenic Inputs", PLoS ONE, 2014

15    28 words / < 1% match - Crossref
N. GYPENS. "Effect of eutrophication on air-sea CO$_2$ fluxes in the coastal Southern North Sea: a model study of the past 50 years", Global Change Biology, 04/2009

16    22 words / < 1% match - Crossref
H.E. Laika. "Interannual properties of the CO2 system in the Southern Ocean south of Australia", Antarctic Science, 08/12/2009

17    21 words / < 1% match - Internet from 08-Jan-2016 12:00AM
www.biogeosciences.net

18    21 words / < 1% match - Crossref
SHIN-ICHIRO NAKAOKA. "Temporal and spatial variations of oceanic pCO2 and air-sea CO2 flux in the Greenland Sea and the Barents Sea", Tellus B, 4/2006

19    20 words / < 1% match - Crossref
Alfonso Mucci. "CO$_2$ fluxes across the air-sea interface in the southeastern Beaufort Sea: Ice-free period", Journal of Geophysical Research, 04/01/2010

| 20 | 17 words / < 1% match - Crossref |
|---|---|
| | Mercedes Paz. "Seasonal variability of surface fCO2 in the Strait of Gibraltar", Aquatic Sciences, 03/2009 |

| 21 | 16 words / < 1% match - Crossref |
|---|---|
| | ANTOINE CORBIÈRE. "Interannual and decadal variability of the oceanic carbon sink in the North Atlantic subpolar gyre", Tellus B, 4/2007 |

| 22 | 14 words / < 1% match - Crossref |
|---|---|
| | Midorikawa, T.. "Estimation of seasonal net community production and air-sea CO"2 flux based on the carbon budget above the temperature minimum layer in the western subarctic North Pacific", Deep-Sea Research Part I, 200202 |

| 23 | 14 words / < 1% match - Crossref |
|---|---|
| | Wang, Xiujun, Raghu Murtugudde, Eric Hackert, Jing Wang, and Jim Beauchamp. "Seasonal to decadal variations of sea surface pCO2 and sea-air CO2 flux in the equatorial oceans over 1984-2013: A basin-scale comparison of the Pacific and Atlantic Oceans : CO2 flux in the equatorial oceans", Global Biogeochemical Cycles, 2015. |

| 24 | 13 words / < 1% match - Crossref |
|---|---|
| | Quay, P.. "Surface layer carbon budget for the subtropical N. Pacific: @d^1^3C constraints at station ALOHA", Deep-Sea Research Part I, 200309 |

| 25 | 12 words / < 1% match - Crossref |
|---|---|
| | Semiletov, I.P.. "Carbonate chemistry dynamics and carbon dioxide fluxes across the atmosphere-ice-water interfaces in the Arctic Ocean: Pacific sector of the Arctic", Journal of Marine Systems, 200706 |

| 26 | 10 words / < 1% match - Internet from 18-Feb-2017 12:00AM |
|---|---|
| | hal.upmc.fr |

| 27 | 10 words / < 1% match - Crossref |
|---|---|
| | Muller, B.. "Influence of organic carbon decomposition on calcite dissolution in surficial sediments of a freshwater lake", Water Research, 200311 |

| 28 | 9 words / < 1% match - Crossref |
|---|---|
| | Gazeau, F.. "The European coastal zone: characterization and first assessment of ecosystem metabolism", Estuarine, Coastal and Shelf Science, 200408 |

| 29 | 9 words / < 1% match - Crossref |
|---|---|
| | C. Dumousseaud. "Contrasting effects of temperature and winter mixing on the seasonal and inter-annual variability of the carbonate system in the Northeast Atlantic Ocean", Biogeosciences, 05/11/2010 |

| 30 | 9 words / < 1% match - Crossref |
|---|---|
| | Reisdorph, S. C., and J. T. Mathis. "Assessing net community production in a glaciated Alaska fjord", Biogeosciences Discussions, 2014. |

| 31 | 8 words / < 1% match - Crossref |
|---|---|
| | Olsen, A.. "Interannual variability in the wintertime air-sea flux of carbon dioxide in the northern North Atlantic, 1981-2001", Deep-Sea Research Part I, 200310/11 |

| 32 | 8 words / < 1% match - Crossref |
|---|---|
| | Xiaomeng Wang. "Late autumn to spring changes in the inorganic and organic carbon dissolved in the water column at Scholaert Channel, West Antarctica", Antarctic Science, 11/24/2009 |

33    8 words / < 1% match - Crossref
Murata, A.. "Summertime CO"2 sinks in shelf and slope waters of the western Arctic Ocean", Continental Shelf Research, 200305

34    8 words / < 1% match - Crossref
Zhang, Jia-Zhong, and Charles J. Fischer. "Carbon dynamics of Florida Bay: spatiotemporal patterns and biological control", Environmental Science & Technology, 2014.

35    7 words / < 1% match - Crossref
Chen, C.-T. A., T.-H. Huang, Y.-C. Chen, Y. Bai, X. He, and Y. Kang. "*Review article* "Air-sea exchanges of CO$_2$ in world's coastal seas"", Biogeosciences Discussions, 2013.

36    6 words / < 1% match - Crossref
Shim, J.. "Seasonal variations in pCO"2 and its controlling factors in surface seawater of the northern East China Sea", Continental Shelf Research, 20071201

37    6 words / < 1% match - Crossref
Hull, T., N. Greenwood, J. Kaiser, and M. Johnson. "Uncertainty and sensitivity in optode-based shelf-sea net community production estimates", Biogeosciences Discussions, 2015.

38    6 words / < 1% match - Crossref
Huertas, I.E.. "Temporal patterns of carbon dioxide in relation to hydrological conditions and primary production in the northeastern shelf of the Gulf of Cadiz (SW Spain)", Deep-Sea Research Part II, 200606

39    6 words / < 1% match - Crossref
N. R. Bates. "Air-sea CO$_2$ fluxes on the Bering Sea shelf", Biogeosciences Discussions, 10/05/2010